# Analyzing machupo virus-receptor binding by molecular dynamics simulations

Austin G. Meyer[1,2,3], Sara L. Sawyer[2], Andrew D. Ellington[2] and Claus O. Wilke[1]

[1] Department of Integrative Biology, Institute for Cellular and Molecular Biology, and Center for Computational Biology and Bioinformatics, The University of Texas at Austin, Austin, TX, USA
[2] Department of Molecular Biosciences, Institute for Cellular and Molecular Biology, The University of Texas at Austin, Austin, TX, USA
[3] School of Medicine, Texas Tech University Health Sciences Center, Lubbock, TX, USA

Corresponding author
Austin G. Meyer,
austin.meyer@utexas.edu

## ABSTRACT

In many biological applications, we would like to be able to computationally predict mutational effects on affinity in protein–protein interactions. However, many commonly used methods to predict these effects perform poorly in important test cases. In particular, the effects of multiple mutations, non alanine substitutions, and flexible loops are difficult to predict with available tools and protocols. We present here an existing method applied in a novel way to a new test case; we interrogate affinity differences resulting from mutations in a host–virus protein–protein interface. We use steered molecular dynamics (SMD) to computationally pull the machupo virus (MACV) spike glycoprotein (GP1) away from the human transferrin receptor (hTfR1). We then approximate affinity using the maximum applied force of separation and the area under the force-versus-distance curve. We find, even without the rigor and planning required for free energy calculations, that these quantities can provide novel biophysical insight into the GP1/hTfR1 interaction. First, with no prior knowledge of the system we can differentiate among wild type and mutant complexes. Moreover, we show that this simple SMD scheme correlates well with relative free energy differences computed via free energy perturbation. Second, although the static co-crystal structure shows two large hydrogen-bonding networks in the GP1/hTfR1 interface, our simulations indicate that one of them may not be important for tight binding. Third, one viral site known to be critical for infection may mark an important evolutionary suppressor site for infection-resistant hTfR1 mutants. Finally, our approach provides a framework to compare the effects of multiple mutations, individually and jointly, on protein–protein interactions.

## INTRODUCTION

The computational prediction of mutational effects on protein–protein interactions remains a challenging problem. Several methods are available to perform an energy difference calculation from an experimentally determined co-crystal structure. For

example, end point methods can be performed rapidly, with relatively low computational cost (*Gront et al., 2011*; *Kortemme, Kim & Baker, 2004*). However, such methods can suffer from various simplifying assumptions. For example, they generally use an implicit solvent approximation and assume the end state difference with minimal structural rearrangement is sufficient to discriminate energetic differences (*Gront et al., 2011*; *Kortemme, Kim & Baker, 2004*). Alternative approaches have been developed using machine learning, training coefficients in a weighted equation containing geometric and energetic parameters (*Vreven, Hwang & Weng, 2011*; *Vreven et al., 2012*; *Bajaj, Chowdhury & Siddahanavalli, 2011*; *Hwang et al., 2010*). Unfortunately, such machine-learning approaches often suffer in novel applications, for which available training sets are small or non-existent. As such, these methods are poorly suited for most host–virus protein–protein systems. By contrast, first principles methods can forgo training, but currently available methods such as free energy perturbation (FEP) and thermodynamic integration (TI) rely on a transitional model (where one state may be wild-type and the other may be a mutant) to make rigorous free energy calculations (*Gilson et al., 1997*; *Lu, Kofke & Woolf, 2004*; *Chodera et al., 2011*; *Gumbart, Roux & Chipot, 2013a*). While these may be considered two of the gold standard techniques for calculating affinity differences, there are a huge number of theoretical and technical complexities that must all be properly managed to ensure a converged solution (*Gumbart, Roux & Chipot, 2013b*). Such considerations quickly come to dominate the protocol, and the necessary bookkeeping introduces the possibility of human error (*Gumbart, Roux & Chipot, 2013b*). Moreover, as the two ending states look ever more dissimilar the chances of convergence fall rapidly. To ensure convergence, these techniques are typically limited to small differences (such as point mutant comparisons) with a few, very impressive exceptions (*Wang, Deng & Roux, 2006*; *Gumbart, Roux & Chipot, 2013a*; *Gumbart, Roux & Chipot, 2013b*). For most investigators, larger differences quickly become intractable as the number of intermediate steps required to compute a converged solution grows or the complexity of adding restraining potentials and computing approximations expands (*Wang, Deng & Roux, 2006*; *Gumbart, Roux & Chipot, 2013a*; *Gumbart, Roux & Chipot, 2013b*).

Here we propose that much of these complexities can be avoided if all we are interested in is a relative comparison of the effects of different mutations on protein–protein interactions, rather than measuring an absolute or relative binding affinity with experimentally realistic units. We impart a pulling force within an all-atom molecular dynamics simulation on one member of the complex while the other is held in place. Then, we measure the force required for dissociation (*Lu & Schulten, 1999*; *Isralewitz, Gao & Schulten, 2001*; *Isralewitz et al., 2001*; *Park & Schulten, 2004*; *Gumbart et al., 2012*; *Miño, Baez & Gutierrez, 2013*). Although such biasing techniques are commonly used in protein-ligand binding problems, they are less commonly applied to protein–protein interactions, and almost never to mutational analysis in a protein–protein system. This is largely the result of free energy convergence difficulties and computational limitations (*Cuendet & Michielin, 2008*; *Cuendet & Zoete, 2011*). Using a proxy for relative binding affinity rather than caluclating

absolute affinities can solve these problems. Here, as proxies, we use the maximum applied force required for separation and the area under the force-versus-distance curve (AUC). For comparison, we also calculate relative free energy differences using the traditional dual topology FEP paradigm, and we show that the two approaches yield congruent results.

We used SMD and FEP to interrogate the interaction between machupo virus (MACV) spike glycoprotein (GP1) and the human transferrin receptor (hTfR1) (*Abraham et al., 2010*; *Charrel & de Lamballerie, 2003*). Machupo virus is an ambisense RNA virus of the arenavirus family (*Charrel & de Lamballerie, 2003*). Worldwide, arenaviruses represent a significant source of emerging zoonotic diseases for the human population (*Charrel & de Lamballerie, 2003*). Members of the arenavirus family include the Lassa fever virus endemic to West Africa, the lymphochoriomeningitis virus (LCMV) endemic to rodents in several areas of the United States, and the Guanarito, Junin, and Machupo viruses endemic to rodents in South America (*Charrel & de Lamballerie, 2003*). The South American arenaviruses typically infect humans after rodent contamination and can cause a devastating hemorrhagic fever with high mortality (*Charrel & de Lamballerie, 2003*).

The hTfR1 is the primary receptor used by MACV for binding its host cell prior to infection. The primary role of hTfR1 *in vivo* is to bind transferrin for cellular iron uptake. The hTfR1 protein contains three extracellular domains: two basilar domains and an apical domain. The two basilar domains serve most of the transferrin-binding function (*Abraham et al., 2010*; *Radoshitsky et al., 2011*). Viral entry is initiated by GP1 binding to the apical domain of hTfR1. Previous work has indicated that the GP1/hTfR1 binding interaction is the primary determinant of MACV host range variation (*Choe et al., 2011*; *Radoshitsky et al., 2011*). The co-crystal structure shows that the high affinity interaction between GP1 and hTfR1 forces the normally flexible loop in the apical domain of hTfR1 into a rigid $\beta$-pleated sheet domain. For GP1, several extended loops mediate binding to hTfR1 (*Abraham et al., 2010*; *Radoshitsky et al., 2011*), and many of the interface interactions are mediated by extensive hydrogen-bonding networks (*Abraham et al., 2010*). Experimental alanine-scanning and whole-cell infectivity assays have identified several sites in both GP1 and hTfR1 that are probably critical for establishing infection (*Choe et al., 2011*; *Radoshitsky et al., 2011*).

We applied our computational method to wild type (WT) and mutant complexes, and found that we could resolve relative differences in unbinding and predict significant affinity changes. Importantly, the affinity changes predicted using only max force or AUC show a strong correlation with rigorous relative free energy differences computed by FEP. At sites known to be important for successful viral entry, we found that the biochemical cause of reduced infectivity may not be as simple as the static structure suggests. For example, the static structure shows a hydrogen-bonding network connected to site N348 in hTfR1. According to our simulations, this network may not affect binding affinity directly. In addition, our study offers an all-atom steered molecular dynamic approach to avoid some of the pitfalls of several existing methods used to evaluate mutations in protein–protein interfaces.

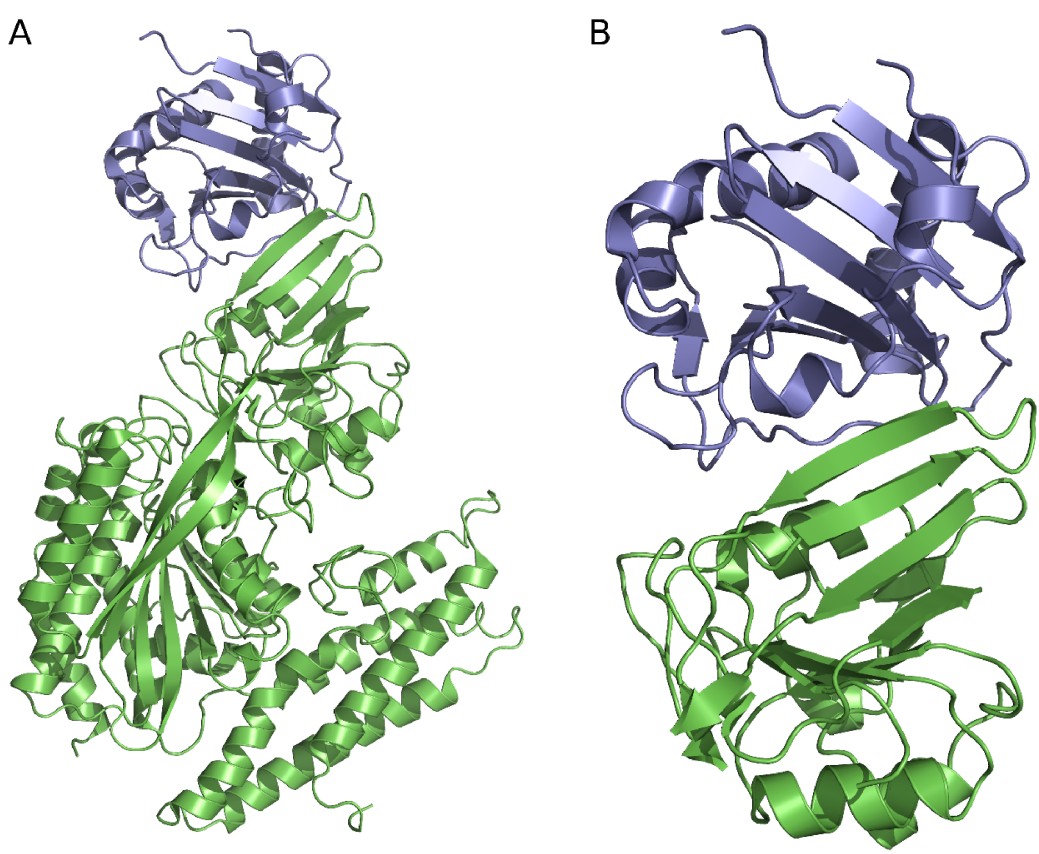

**Figure 1** **The GP1/hTfR1 complex.** GP1 is shown in blue and hTfR1 is shown in green. (A) The full, de-glycosylated GP1/hTfR1 co-crystal structure. (B) The reduced structure used in SMD simulations.

# MATERIALS AND METHODS

## System modeling

For our experiments, we used the experimentally determined GP1/hTfR1 structure (PDB-ID: 3KAS) (*Abraham et al., 2010*). The apical domain of hTfR1 interacts directly with GP1 while the other two domains are closer to the cell membrane and have essentially no interaction with GP1. The biophysical independence of the apical domain allowed us to isolate it without significantly affecting the GP1/hTfR1 interaction.

We used the protein visualization software PyMOL (*Schrödinger, 2010*) to remove residues 121–190, 301–329, and 383–756 in the hTfR1. No residues were removed from the viral protein. Figure 1 shows a model of the initial structure and that of the pared structure. Although GP1 has several glycosylatable residues, we opted to use the de-glycosylated protein for this study. The complexity of correctly parameterizing diverse sugar moieties is outside of the scope of this paper. Furthermore, although it is known that GP1 is glycosylated, and some of those sugars contact hTfR1, the sugars in the available PDB structure are not physiological for mammals (*Abraham et al., 2010*). In total we removed 10 sugars from the crystal structure for this study.

After system reduction, the Visual Molecular Dynamics (VMD) (*Humphrey, Dalke & Schulten, 1996*) package along with its system of back-ends was used for all subsequent modeling. The Orient add-on package allowed us to rotate the system axis such that the direction of steering was oriented directly down the *z*-axis. De-glycosylation simplified the system such that Autopsf could easily find the chain terminations and patch them appropriately. The Solvate package was used to generate a TIP3P water model with a 5 Å buffer (relative to the maximum dimensions of the proteins) on all sides except down the positive *z*-axis where a 20 Å buffer was created. Finally, we used the Autoionize package to place 150 millimolar NaCl and neutralize the total system charge. In the end, each modeled system had approximately 28,000 atoms.

## Equilibration

NAMD was used for all simulations in this study (*Phillips et al., 2005*). In addition to the modeled system, for equilibration we generated a configuration file that fixed the $\alpha$-carbon backbone. This was accomplished by setting the B-factor column to 1 for the fixed atoms and to zero for all other atoms. Further, we generated a configuration file with fixed $\alpha$-carbon atoms at residues 41–92 (numbered linearly, in this case, starting at 1 for the first amino acid as was required for NAMD) in the hTfR1. The second file was used to affix a harmonic restraint, thus preventing any unfolding due to system reduction. More importantly, the harmonic restraint allowed the protein complex to equilibrate while preventing any drift from its predefined position; the restraint did not constrain the structure of each protein, or the relative position or orientation of the two proteins to each other. Finally, we calculated the system center and dimensions for use in molecular dynamics settings. The exact NAMD configuration files are available on GitHub (https://github.com/clauswilke/MACV_SMD).

We used the Charmm27 (*Brooks et al., 1983*) all-atom force field. The initial system temperature was set to 310 K. Several typical MD settings were used including switching and cutoff distances (see provided configuration files). In addition, we used a 2 femtosecond time step with rigid bonds. We used periodic boundary conditions with the particle mesh ewald (PME) method of computing full system electrostatics outside of the explicit box. Furthermore, we used a group pressure cell, flexible box, langevin barostat, and langevin thermostat during equilibration. A harmonic restraint (called harmonic constraint in VMD) was set as stated previously.

To start the simulation, the barostat was switched off and the system was minimized for 1000 steps. Next, the fixed backbone was released, and the system was minimized for an additional 1000 time steps. Subsequently, the system was released into all-atom molecular dynamics for 3000 steps. Finally, the langevin barostat was turned on and the system was simulated for 2 ns (1,000,000 steps) of chemical time. For each mutant, twenty independent equilibration replicates were run with an identical protocol.

## Steered molecular dynamics

We used the final state from each equilibrated system to restart another MD simulation. Our steering protocol is fundamentally similar to *Cuendet & Michielin (2008)* with slightly

different parameter choices. Perhaps the one significant difference lies in our choosing to not use a thermostat or barostat. We can make this choice because we are not trying to calculate the binding free energy by any physically rigorous approach (the Jarzynski inequality being one example). Following equilibration, the final state of each simulation was used to generate a configuration file fixing the $\alpha$-carbon on residues 1, 58, 73–83, 96, 136, 137, 138, and 161 (again with linear numbering) in the hTfR1. These residues were selected as they are far from the binding interface and sufficiently distributed to prevent any orientational motion of the receptor relative to the viral spike protein. The center of mass of the $\alpha$-carbons of all residues (163–318 in linear numbering) in GP1 received an applied force during the simulation. The NAMD convention does not actually apply a force to all $\alpha$-carbon atoms but rather uses the selection to compute an initial center of mass. Then, during the steering run, the single center of mass point is pulled with the parameters described below. We used the same force field parameters (exclude, cutoff, switching, etc.), the same integrator parameters (time step, rigidbonds on, all molecular being wrapped, etc.), and the same particle mesh ewald parameters as in equilibration. Periodic boundary conditions were incorporated as part of the system (as is the convention in NAMD restart) and PME was again used to approximate full system electrostatics.

We ran test simulations at several force constants and visually inspected the results. A force constant of 5 kcal/mol/Å$^2$ was chosen due to its relatively low signal-to-noise ratio. This constant is slightly lower than the more common 7 kcal/mol/Å$^2$ found in several recent studies; that value is commonly selected primarily because it is the force constant found in the SMD tutorial available through the NAMD developers. Moreover, the force constant could very likely be set to a range of nearby values with little loss in predictive power.

In SMD experiments the pulling velocity should be as low as possible for the available computational time (*Cuendet & Michielin, 2008*; *Cuendet & Zoete, 2011*). We choose a velocity of 0.000001 Å/fs = 1 Å/ns, and direction down the positive $z$-axis. One could use faster pulling if the computing time must be reduced, but slower than necessary pulling speeds are not typically considered problematic.

SMD was run for 15 ns (7,500,000 time steps) of chemical time. For each simulation, we randomly selected one of the equilibration runs for restart. We ran 50 replicate simulations per mutant for a total of 550 SMD simulations. All GP1/hTfR1 complexes separated by greater than 4 Å and many separated to 10 or more.

To leave the final trajectory of a tractable size, only 1000 evenly spaced frames were retained from each simulation, leaving a final trajectory size of 323 MB. See Movie S1 for a representative unbinding trajectory. Initial development of the SMD protocol was carried out on the Lonestar cluster at the Texas Advanced Computing Center (TACC). All production SMD simulations were performed on the Hrothgar cluster at Texas Tech University, using NAMD 2.9. Each simulation was parallelized over 60 computational cores and utilized approximately 20 h of computing time. The total chemical time simulated for this project was nearly 10 µs, requiring slightly over 1 million cpu-hours.

## Free energy perturbation

Briefly, we used the traditional dual topology approach to FEP (*Gao et al., 1989*; *Pearlman, 1989*). This involves a thermodynamic cycle where a set of atoms are progressively decoupled from the environment while another set of atoms are progressively coupled. To compute the relative free energy difference requires knowing the free energy change when the transformation is carried out for the bound complex and the individual protein. Then, one can compute the relative free energy difference between a WT and mutant complex by taking the difference between the energy required to decouple/couple the atoms in solution from the energy required to decouple/couple the atoms in the bound complex (*Gao et al., 1989*; *Pearlman, 1989*).

Again, the NAMD configuration file is made available via GitHub (https://github.com/clauswilke/MACV_SMD). We used a similar configuration to that in equilibration. One significant difference was to make a cubic water box with a side length equal to the long axis of the complex plus a 10 Å buffer on either side, and simply restrict center of mass motion with the NAMD setting. This was done to avoid affecting the system energy while calculating free energy differences.

The transition protocol for bound and free protein systems were identical. They started with 1000 steps of minimization and 250,000 steps of equilibration in the starting state for the forward and reverse directions. Phase transitions were carried out in steps of $\lambda = 0.05$. Each transition was carried out for 250,000 steps. The first 100,000 steps after phase transition were reserved for equilibration and the final 150,000 steps were used for data collection.

The VMD mutator tool was used to generate the necessary topology file and the parseFEP tool (*Liu et al., 2012*) in VMD was used for subsequent analysis. We used it to perform error analysis and compute the Bennett acceptance ratio as the maximum likelihood free energy difference of the two states under consideration. Though the larger transitions presented difficulty in a small number of windows, forward and reverse hysteresis was generally in good agreement for all complexes. The double mutants were performed by first doing the Y211A mutation followed by the other of the two mutants. Then, the $\Delta G$'s were simply added together to get the total energetic difference.

## Post-processing

The python packages MDAnalysis (*Michaud-Agrawal et al., 2011*) and ProDy (*Bakan, Meireles & Bahar, 2011*) were both used at various points in post-processing. The molecular trajectory (comprising the atomic coordinates per time) was parsed to compute the center-of-mass for each of the two complexes. The starting center-of-mass distance was set to zero and the distance was re-computed at each time step relative to the starting distance.

The statistical package R was used for all further analysis and visualization. Each of the 50 independent trajectories per mutant produced a fairly noisy force curve. The force curves for each mutant were smoothed over all replicates by using the smooth.spline() and predict() functions in R with default settings. The two primary descriptive statistics

we used were maximum interpolated applied force and total area under the interpolated curve (AUC). We tested for signfiant differences in maximum force or AUC by carrying out t tests for all pairwise combinations (each mutant compared to each other mutant), using the pairwise.t.test() function in R. We adjusted $p$ values to correct for multiple testing using the False-Discovery-Rate (FDR) method (*Benjamini & Hochberg, 1995*). The ggplot (*Wickham, 2009*) package was used to generate most of the figures.

Analysis scripts and final data (except MD trajectories) are available on the github repository accompanying this publication (https://github.com/clauswilke/MACV_SMD).

## RESULTS

### The GP1/hTfR1 system

The GP1/hTfR1 interface (Fig. 2) marks a particularly important and useful test system. There are several sites on both the human and viral protein known to affect the infectivity phenotype of MACV. Many of the important sites have been mapped by *in vitro* flow-cytometry based entry assays. The GP1/hTfR1 interface appears not to be dominated by one particular type of interaction (electrostatics, hydrogen-bonding, or van der Waals). In addition, much of the binding domain on hTfR1 is on a loop that is flexible prior to viral binding, but organizes to become a strand of a $\beta$-sheet on binding. As a result, many other computational techniques (*Gront et al., 2011*; *Kortemme, Kim & Baker, 2004*) are only marginally useful. The complex nature of this interface represents a particularly difficult challenge for traditional computational analysis.

In total, we tested 7 point mutants and 3 double mutants in addition to the WT complex (Table 1). All of the mutations are within 5 Å of the protein–protein interface. Mutations in hTfR1 at site 211 have proven capable of causing loss-of-entry according to *in vitro* flow-cytometry infection assays or known host-range limitations (*Radoshitsky et al., 2008*; *Choe et al., 2011*; *Radoshitsky et al., 2011*). Most likely, this effect is caused by the destruction of a critical hydrogen bond to Ser113 or Ser111 in GP1. The lost hydrogen bond would lead to the subsequent loss of a large hydrogen-bonding network seen in the crystal structure (Table 1) (*Abraham et al., 2010*). In a manner similar to site 211, N348 appears to be important for binding by participating in a critical hydrogen bonding network (*Radoshitsky et al., 2008*; *Abraham et al., 2010*) to GP1. In particular, N348Lys is reported in the literature to cause significantly reduced viral entry *in vivo* (Table 1) (*Radoshitsky et al., 2008*; *Abraham et al., 2010*). Finally, an alanine mutation at site 111 in GP1 (mutation vR111A) has also been shown to cause decreased entry (Table 1) (*Radoshitsky et al., 2011*). For notation purposes, the viral site is always referred to with a preceding 'v'.

Despite the fact that viral binding occurs at the site of a flexible loop in the free hTfR structure, our data shows after binding the strand is extremely rigid. In the bound conformation, only two sites of the loop have root mean squared fluctuation (RMSF) values in the top half of all receptor sites during equilibration (Fig. 3), and those are almost completely exposed to solvent. This is unsurprising considering the high degree of burial that occurs as a result of viral binding. Computing the root mean squared deviation

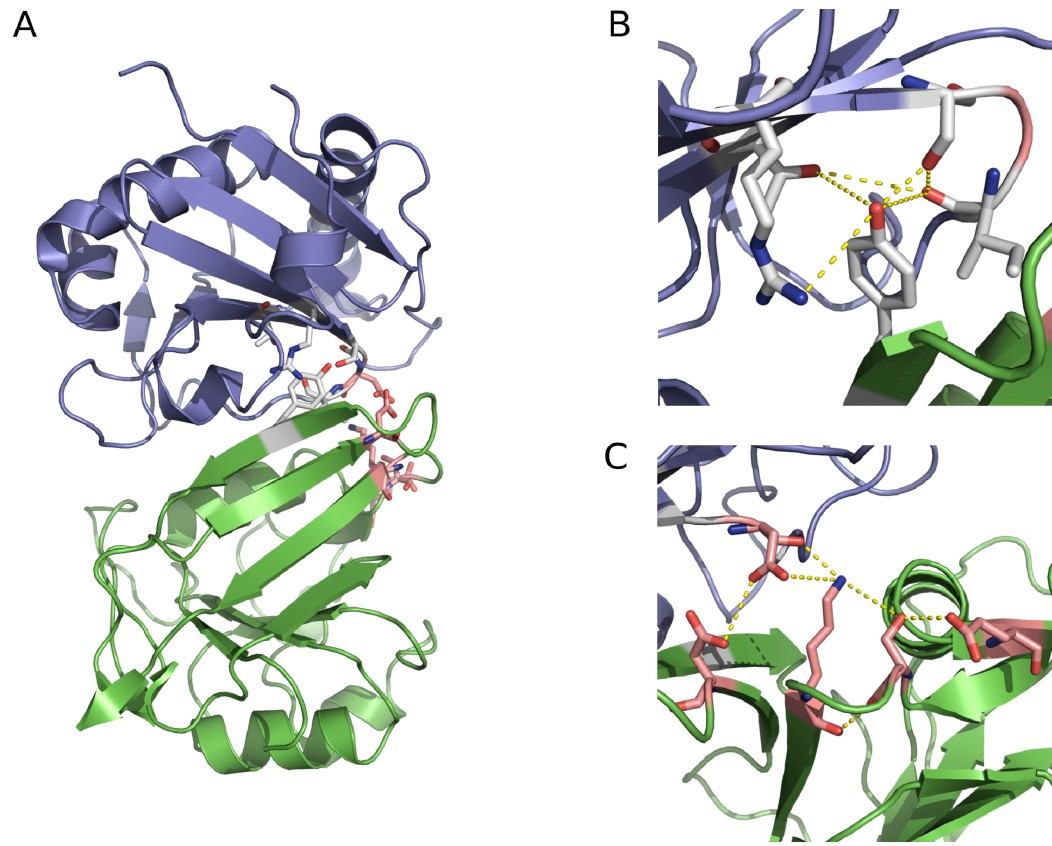

**Figure 2** **The two hydrogen bonding networks.** GP1 is shown in blue and hTfR1 is shown in green. (A) The first network including Y211 and R111 is shown in white, and the second network containing N348 is shown in pink. (B) Near view of the first network with contacts in yellow. (C) Near view of the second network with contacts in yellow.

**Table 1** **Summary of prior information available for each mutation tested.** Observed *in vivo* refers to mutations that have been observed in rodent populations. Phenotype *in vitro* refers to the observed phenotype in *in vitro* viral entry assays.

| Mutation | Observed *in vivo* | Phenotype *in vitro* |
| --- | --- | --- |
| WT | Yes | Normal entry |
| N348A | No | – |
| N348K | Yes | Diminished entry |
| N348W | No | – |
| vR111A | No | Diminished entry |
| N348A/Y211A | No | – |
| vR111A/Y211A | No | – |
| Y211D | Yes | No expression |
| Y211T | No | Diminished entry |
| Y211A | No | No expression |
| N348W/Y211A | No | – |

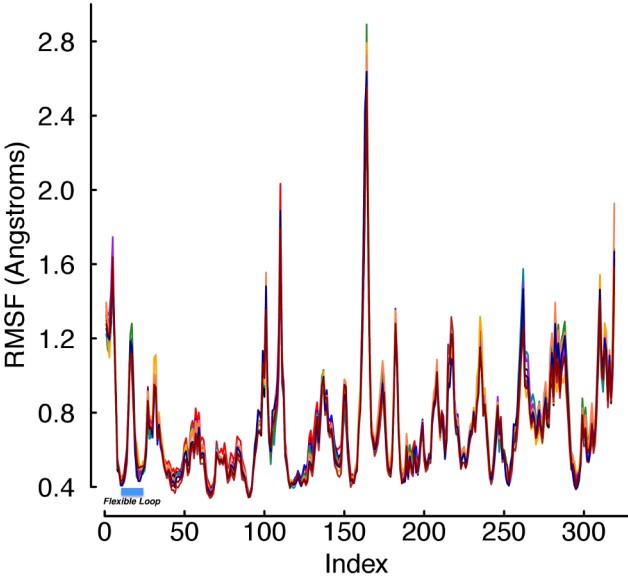

**Figure 3 RMSF values during equilibration.** The RMSF values for every site in the bound complex computed during the equilibration phase of the protocol. Each color represents the average over 20 trajectories of a single mutant. Indices 17–25 are the hTfR flexible loop. The plot shows the flexibility of each site is essentially independent of mutation, and two sites (indices 17 and 18) are a part of the flexible loop in the free receptor. However, these two residues are not above 0.72 Å actually found in the protein–protein interface, but rather are almost completely solvent exposed with the virus bound.

(RMSD) of the entire structure over the trajectory shows that none of the mutations are so deleterious as to cause rapid unbinding. In fact, the RMSD over trajectory looks highly invariant across mutants (Fig. 4). In the unbound state, calculated near the end of the SMD trajectory, all of the residues in the WT receptor interfacial strand are in the top half of RMSF over all receptor sites (Fig. 5). Thus, if sufficient simulation time is not dedicated to allowing this unfolding process, standard free energy techniques may miss the energetic contributions that result from ordering the flexible loop in the hTfR apical domain.

## Molecular dynamics simulations

We analyzed the GP1/hTfR1 system using two molecular dynamics techniques. First, by carrying out SMD using a known force constant and pulling with a constant velocity, we could calculate the applied force during protein–protein dissociation (*Cuendet & Michielin, 2008*; *Cuendet & Zoete, 2011*). A typical averaged force curve comparison can be seen in Fig. 6, and individual images of all averaged force curves are available in the associated GitHub repository, in folder figures/force_curves. As seen in Fig. 6, the dissociation distance was relatively consistent among mutants. Movie S1 visually illustrates the separation distance between peptide domains. The quantities maximum applied force and AUC were derived from the force-versus-distances curves. Their summary statistics are reported in Table 2. As we are more interested in the phenotypic impact of interface mutations we avoided many of the more physically rigorous, but technically complicated calculations that are possible with SMD (*Isralewitz, Gao & Schulten, 2001*; *Isralewitz et al., 2001*).

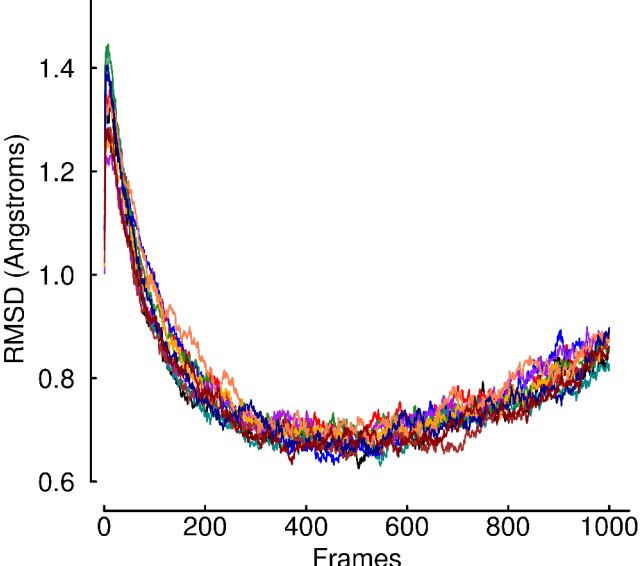

**Figure 4  RMSD values during equilibration.** The RMSD values over the time of the trajectory computed during the equilibration phase of the protocol. Each color represents the average over 20 trajectories of a single mutant. The plot shows none of the mutants causes immediate unbinding of the protein–protein complex. In addition, the universal upward trend near the end of the equilibration trajectories may indicate the crystal is more tightly packed than would normally occur in solution.

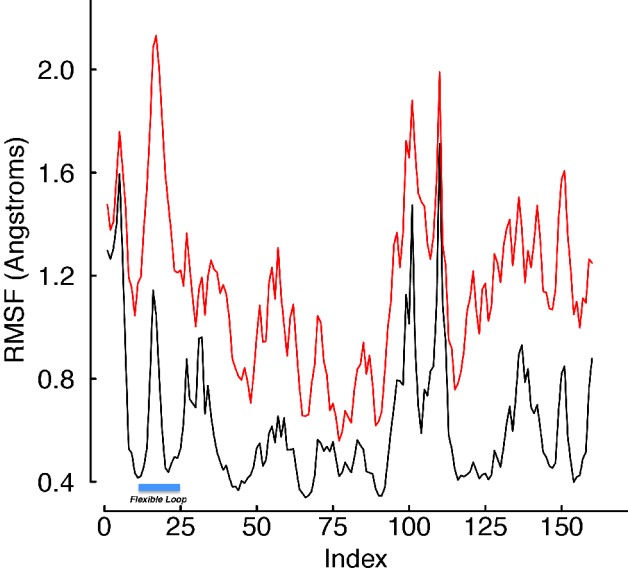

**Figure 5  RMSF values of WT hTfR in equilibration and SMD.** The RMSF values for every site in the WT receptor were computed during the equilibration phase and during final 50 frames of the SMD trajectories. The black line was computed over equilibration and the red line during SMD. The plot shows the solution mobility of the hTfR flexible loop increases more than the average during the unbinding process.

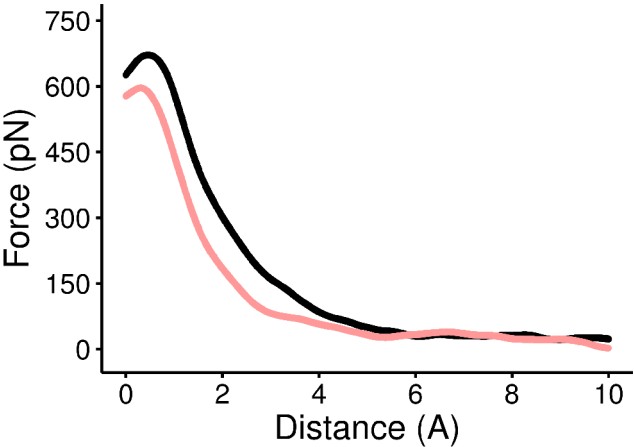

**Figure 6 Force versus distance curve of WT and the Y211A mutant.** The average force curve for 50 replicates of the WT complex is shown in black, and the average of 50 replicates of the Y211A mutant is shown in red. There is a large difference in both maximum applied force and AUC between the two complexes.

**Table 2 Summary statistics for each mutation tested.** $\mu_{MAF}$ is the mean in piconewtons and $\sigma_{MAF}$ is the standard deviation of maximum applied force over all simulations. $\mu_{AUC}$ is the mean and $\sigma_{AUC}$ is the standard deviation of AUC over all simulations. $\Delta G$ is the free energy difference in kcal/mol calculated via FEP by the dual topology paradigm.

| Mutation | $\mu_{MAF}$ (pN) | $\sigma_{MAF}$ | $\mu_{AUC}$ | $\sigma_{AUC}$ | $\Delta G$ (kcal/mol) |
|---|---|---|---|---|---|
| WT | 734.4856 | 131.6513 | 145460.4 | 60232.26 | 0.000 |
| N348A | 748.5217 | 137.4864 | 133913.9 | 51078.64 | −2.149 |
| N348K | 705.0707 | 108.5079 | 141084.4 | 54450.28 | +3.184 |
| N348W | 697.3642 | 132.6436 | 136886.0 | 53796.44 | +3.033 |
| vR111A | 713.8081 | 106.7374 | 136103.2 | 52070.85 | +0.466 |
| N348A/Y211A | 703.7027 | 128.5866 | 113464.2 | 57451.62 | +5.203 |
| vR111A/Y211A | 741.0642 | 131.6287 | 130070.6 | 47665.56 | −2.440 |
| Y211D | 825.2586 | 115.4343 | 158878.7 | 63039.08 | −2.760 |
| Y211T | 806.8593 | 136.5648 | 167110.7 | 78849.29 | +0.875 |
| Y211A | 654.1138 | 108.5343 | 108090.0 | 43661.09 | +2.526 |
| N348W/Y211A | 594.9044 | 134.8233 | 108984.2 | 45451.00 | +8.206 |

Before systematically applying SMD to the GP1/hTfR1 interaction, we needed to ensure the method was sufficiently sensitive to distinguish between relatively minor point mutations. While SMD has been applied previously to measure the binding energy of high-affinity T-cell receptor interactions (*Cuendet & Michielin, 2008*; *Cuendet & Zoete, 2011*), it is rarely used to parse small energy differences in a protein–protein interaction energy landscape. For this initial sensitivity analysis, we tested alanine substitutions congruent with the traditional experimental and computational approach.

We proceeded to compare our SMD results to that of the standard dual topology FEP approach to calculate relative free energy differences. The correlation between the

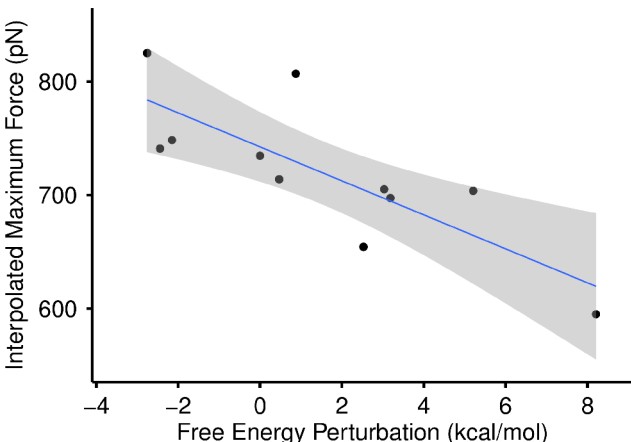

**Figure 7 Max force versus free energy perturbation.** Scatter plot of maximum force in SMD versus the relative free energy difference calculated by FEP for all 10 mutants tested plus the WT complex. The WT complex for FEP was simply set to 0.0. The correlation between the two is $r = -0.795$ with $p = 0.0034$.

**Table 3 Pairwise differences (row variable minus column variable) in mean maximum applied force.** Bolded values are statistically significant at $p < 0.05$.

| | WT | N348A | N348W | N348K | vR111A | N348A/Y211A | vR111A/Y211A | Y211D | Y211T | Y211A |
|---|---|---|---|---|---|---|---|---|---|---|
| N348A | +14.036 | | | | | | | | | |
| N348W | −29.414 | −43.451 | | | | | | | | |
| N348K | −37.121 | **−51.157** | −7.7060 | | | | | | | |
| vR111A | −20.677 | −34.713 | +8.7370 | +16.443 | | | | | | |
| N348A/Y211A | −30.782 | −44.819 | −1.3670 | +6.3380 | −10.105 | | | | | |
| vR111A/Y211A | +6.5790 | −7.4570 | +35.993 | +43.700 | +27.256 | +37.361 | | | | |
| Y211D | **+90.772** | **+76.736** | **+120.19** | **+127.89** | **+111.45** | **+121.56** | **+84.194** | | | |
| Y211T | **+72.373** | **+58.337** | **+101.79** | **+109.50** | **+93.051** | **+103.16** | **+65.795** | −18.399 | | |
| Y211A | **−80.371** | **−94.407** | −50.956 | −43.250 | −59.694 | −49.588 | **−86.950** | **−171.1 4** | **−152.75** | |
| N348W/Y211A | **−139.58** | **−153.62** | **−110.17** | +102.46 | −118.903 | −108.80 | +146.16 | +230.35 | **−211.95** | −59.209 |

energetically rigorous FEP and our statistical approach is high. For all 11 complexes tested, the correlation between max force and FEP was $r = -0.795$ at $p = 0.0034$ (Fig. 7), and the correlation between AUC and FEP was $r = -0.593$ at $p = 0.055$. Because of the strong correlation, we refer exclusively to the SMD results for the remainder of this work, focusing primarily on max force.

We found that relative to WT, one alanine mutation (Y211A) produced a very large and statistically significant difference in the maximum applied force and AUC (Fig. 6, Table 3), while the other two did not (Table 3). When considering additional mutants (also discussed below), we found that maximum applied force was generally sufficient to distinguish mutants (Tables 3 and 4), and AUC was able to add a few more statistically significant differences (Table 5). In general, however, and consistent with the FEP results, maximum applied force seemed to be the more sensitive statistic than AUC.

**Table 4 Pairwise difference p-values for maximum applied force.** Bolded values are statistically significant at $p < 0.05$.

| | WT | N348A | N348W | N348K | vR111A | N348A/Y211A | vR111A/Y211A | Y211D | Y211T | Y211A |
|---|---|---|---|---|---|---|---|---|---|---|
| N348A | 0.60 | | | | | | | | | |
| N348W | 0.31 | 0.077 | | | | | | | | |
| N348K | 0.20 | **0.038** | 0.81 | | | | | | | |
| vR111A | 0.51 | 0.16 | 0.79 | 0.60 | | | | | | |
| N348A/Y211A | 0.29 | 0.07 | 0.95 | 0.81 | 0.77 | | | | | |
| vR111A/Y211A | 0.82 | 0.79 | 0.21 | 0.13 | 0.35 | 0.20 | | | | |
| Y211D | **0.00093** | **0.0012** | **$1.4 \times 10^{-5}$** | **$5.0 \times 10^{-6}$** | **$5.6 \times 10^{-5}$** | **$1.2 \times 10^{-5}$** | **0.0022** | | | |
| Y211T | **0.01** | **0.018** | **0.00022** | **$8.7 \times 10^{-5}$** | **0.0008** | **0.0002** | **0.021** | 0.56 | | |
| Y211A | **0.0034** | **$7.2 \times 10^{-5}$** | 0.074 | 0.13 | **0.035** | 0.079 | **0.0016** | **$4.2 \times 10^{-10}$** | **$4.2 \times 10^{-8}$** | |
| N348W/Y211A | **$3.9 \times 10^{-7}$** | **$1.1 \times 10^{-10}$** | **$6.5 \times 10^{-5}$** | **0.00021** | **$1.6 \times 10^{-5}$** | **$7.2 \times 10^{-5}$** | **$1.3 \times 10^{-7}$** | **$< 2 \times 10^{-16}$** | **$2.0 \times 10^{-14}$** | **0.036** |

Table 5 **Pairwise difference *p*-values for interpolated AUC.** Bolded values are statistically significant at $p < 0.05$.

| | WT | N348A | N348W | N348K | vR111A | N348A/Y211A | vR111A/Y211A | Y211D | Y211T | Y211A |
|---|---|---|---|---|---|---|---|---|---|---|
| N348A | 0.33 | | | | | | | | | |
| N348W | 0.76 | 0.59 | | | | | | | | |
| N348K | 0.59 | 0.80 | 0.76 | | | | | | | |
| vR111A | 0.55 | 0.85 | 0.76 | 0.94 | | | | | | |
| N348A/Y211A | **0.017** | 0.07 | **0.031** | 0.076 | 0.08 | | | | | |
| vR111A/Y211A | 0.26 | 0.76 | 0.46 | 0.68 | 0.72 | 0.22 | | | | |
| Y211D | 0.33 | **0.029** | 0.18 | 0.09 | 0.08 | **0.00046** | **0.029** | | | |
| Y211T | 0.09 | **0.0056** | **0.046** | **0.027** | **0.023** | **$4.1 \times 10^{-5}$** | **0.006** | 0.59 | | |
| Y211A | **0.0056** | **0.027** | **0.016** | **0.029** | **0.031** | 0.75 | 0.09 | **$8.2 \times 10^{-5}$** | **$8.5 \times 10^{-6}$** | |
| N348W/Y211A | **0.006** | **0.029** | **0.017** | **0.032** | **0.034** | 0.76 | 0.1 | **$9.4 \times 10^{-5}$** | **$8.5 \times 10^{-6}$** | 0.94 |

## Comparative analysis of the GP1/hTfR1 interface

Considering the involvement of extended hydrogen-bonding networks in the GP1/hTfR1 interface (Fig. 2), it was not clear that individual alanine mutations, even those that should destroy such networks, would significantly change the strength of interaction. One major advantage of first principles simulations is the ability to test mutations other than alanine without additional underlying assumptions in the energy function. As shown in Table 1, we made additional mutations based on biochemical intuition or available experimental data to chemically diverse amino acids including tryptophan, lysine, aspartate, and threonine. Several mutations caused significant relative affinity changes. In addition, to detect synergistic effects, we tested several double mutants where both mutations appeared to cause similar changes in binding. Then, we compared the size of those differences to single mutants (Figs. 8 and 9).

Although Y211A appears to have a large impact on binding affinity, no single mutant can provide enough evidence to understand the biochemical difference in binding mechanism. Since alanine is both smaller than tyrosine and also incapable of participating in hydrogen-bond interactions, we tested further mutations to identify the critical biochemical difference responsible for change in binding affinity. In particular, we substituted smaller side chains that, like tyrosine, were capable of hydrogen bonding. We chose Y211D and Y211T, two mutations that have been discussed in the context of selection pressure on hosts in rodent populations (*Radoshitsky et al., 2008*; *Choe et al., 2011*; *Radoshitsky et al., 2011*). Both mutations proved capable of causing a significant change in binding affinity in our simulations, but the change appeared to be increased affinity (Figs. 8 and 9, and Table 4).

We also simulated several point mutations at N348 in the hTfR1. As discussed above, the alanine mutation at this site showed no significant difference in maximum applied force or AUC from WT (Tables 4 and 5). In addition, neither the N348Lys nor the N348W mutation showed a significant difference from WT. For both of these mutations, however, mean maximum applied force and mean AUC was lower than for WT (See Table 2). On the other hand, there was a detectable difference between N348A and N348Lys (Tables 4
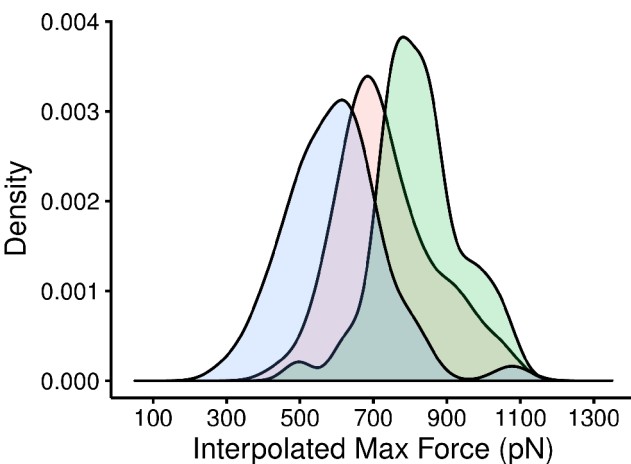

**Figure 8 Distribution of interpolated maximum force for three different GP1/hTfR1 complexes.** The WT GP1-hTfR1 complex in the middle is flanked by the tighter binding mutant Y211D on the right and the weaker binding double mutant N348W/Y211A on the left. The large non-overlapping areas indicate a large and statistically significant difference in these three complexes.

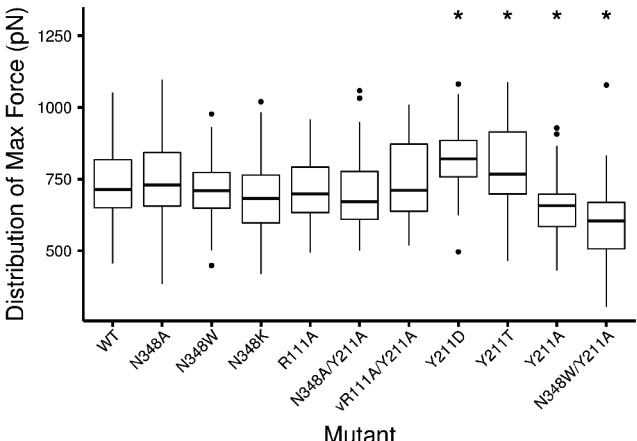

**Figure 9 Distribution of interpolated maximum force for all bound complexes tested.** Stars above the boxplots indicate a statistically significant difference in mean maximum force relative to the WT complex.

and 5), with N348Lys being a weaker binder. Moreover, N348W showed nearly identical results to N348Lys. The mutations to large amino acids (N348W and N348Lys) produced nearly identical affinity changes, whereas the mutations to amino acids not capable of hydrogen bonding (N348A and N348W) produced significantly different affinity changes (Table 3). To check the consistency of our results, we hypothesized that the combination of Y211A and N348W, being chemically disconnected in two different hydrogen-bonding networks, would lead to a synergistic loss-of-binding. As expected, the double mutant was the weakest binding mutant tested ($p < 10^{-6}$, Tables 4 and 5) in this study. Further, according to maximum applied force (but not AUC), the combination of Y211A and N348W also showed significantly weaker binding than Y211A by itself (Tables 4 and 5).

We suspect that the effect of N348W alone is near the limit of detection using our method. A larger number of replicates would possibly have resolved affinity differences between N348W and WT or other mutants more consistently.

Last, we further analyzed a single mutation in GP1, vR111A. As mentioned previously, in our simulations this mutant showed no significant change in either maximum applied force or AUC (Tables 4 and 5), even though both quantities were, on average, lower than in WT (Table 2). This result was somewhat surprising, since Y211A, presumably disrupting the same hydrogen-bonding network as vR111A, displayed a significant reduction in affinity. To probe the interaction between position 111 in the GP1 and position 211 in the hTfR1 further, we also tested the double mutant vR111A/Y211A. This double mutant showed affinity indistinguishable from WT and significantly higher than Y211A alone (Table 3). This result shows that the two sites do indeed interact, and that replacing the hydrogen-bonding network at these sites with a hydrophobic interaction could lead to comparable binding affinity.

## DISCUSSION

We have applied a method utilizing steering forces in all-atom molecular dynamics simulations to evaluate the effects of mutations at the GP1/hTfR1 interface. We modeled mutations at several sites in the GP1/hTfR1 interface, and verified that our computational protocol was sensitive enough to distinguish point mutants in hTfR1. Further, we identified two test statistics, maximum applied force and AUC, that can be used as proxies for binding affinity. Both of these statistics correlate well with FEP, but offer the simplicity of not requiring a large commitment to planning for the theoretical issues inherent to free energy methods. We systematically tested several point mutations to understand their contribution to the binding interaction. In the case of N348Lys, we have shown that the static structure provides little insight into why this mutation causes loss-of-infectivity *in vivo*. While N348 appears to be involved in a hydrogen-bonding network in the static structure, change in binding at that site may actually be caused by size and charge restriction. We also found that a negatively polar residue at site 211 in hTfR1 seem critical for a tight binding interaction. Any non-polar mutation at Y211 in hTfR1 is likely to completely halt viral entry and dramatically decrease the chances of MACV infection.

Traditionally SMD has been either applied to compute equilibrium free energies via a non-equilibrium approximation (*Park et al., 2003*; *Park & Schulten, 2004*; *Giorgino & Fabritiis, 2011*), used to estimate protein stability through unfolding (*Lu & Schulten, 1999*), or used to calculate the absolute free energy of small molecule ligand binding (*Dixit & Chipot, 2001*). Likewise, others have used SMD to understand the process of binding and unbinding at a resolution unmatched by experiment (*Cuendet & Zoete, 2011*; *Giorgino & Fabritiis, 2011*). Here, we have shown that SMD can provide insight into the *relative* strength of protein–protein interactions. Via SMD, one can separate mutations whose likely effect is altered binding affinity with simple statistics like maximum force of separation. Thus, SMD may open avenues for subsequent experimental work in some situations where FEP may be prohibitively difficult.
Our findings rationalize several effects observed in both infectivity data and rodent populations (*Radoshitsky et al., 2008*; *Choe et al., 2011*). First, we found that some substitutions at positions 211 and 348 did affect the strength of receptor binding. However, the computational data suggest that the reason and nature of the effects at these two sites are very different. At position 211, mutations to non-polar residues cause a large change in binding. This is congruent with what is known from viral entry data (*Radoshitsky et al., 2008*; *Choe et al., 2011*). By contrast, mutations at position 348 need only be small to maintain WT binding. The ability to hydrogen bond appears to be insignificant. This can be inferred from the fact that Y211A paired with large (W) and positively charged (Lys) substitutions at position 348 results in a larger than expected synergistic difference. That is, the double mutant Y211A/N348W caused a much larger decrease in binding than we expected from either mutation individually. Third, the GP1 mutation vR111A causes a loss-of-infection during *in vitro* infectivity assays (*Radoshitsky et al., 2011*), yet it was indistinguishable from the WT complex in our simulations. Although Y211A was the most disruptive single mutant we tested, vR111A in the GP1 was able to restore mean maximum applied force to WT levels (Table 2), and to levels significantly higher than observed for Y211A alone.

We would like to emphasize here that we cannot expect perfect agreement between our simulations and the available experimental data, but the correspondence to a well established free energy method bolsters our conclusions. While we have shown that our method can distinguish individual point mutations, we do not know the limit of detection with our method. First, it is possible that some mutants display measurable phenotypic effects in experiments yet appear identical in simulation. More extensive sampling or refinement of the simulation protocol could help to differentiate such mutants (see also next paragraph). Second, the SMD method is fundamentally limited by the accuracy of our starting structure. Third, the available experimental data for the GP1/hTfR1 system were generally obtained from entry assays or whole-cell binding assays rather than molecular binding assays. A mutant may cause a phenotypic difference in infectivity without generating a signal by our method. For example, entry could be lost in the experimental system because the protein is grossly or partially misfolded. An additional analytical step with circular dichroism or an analogous technique could clarify such large-scale folding differences. Further, since our simulations start with a bound structure, any changes that may dramatically affect the rate of association (different folds, trafficking issues, etc.) or relative orientation of the two proteins would be underestimated by our method.

There are a few additional challenges for investigating host–virus interactions via molecular dynamics simulation. As with any atomistic simulation, there is going to be a fairly large noise-to-signal ratio. To reduce noise, one could further customize each simulation, e.g., by determining the optimal pulling speed. Furthermore, larger amounts of computational resources will have a direct and powerful impact on the strength of any atomistic study (*Jensen et al., 2012*). Such resources could come in the form of increased compute time, improved code, or customized hardware for floating point operations (*Shaw et al., 2009*). With improved resources, we could investigate thousands of

individual permutations in the GP1/hTfR1 binding interface. In addition, with additional compute time it would be possible to incorporate equilibrium sampling approaches (*Buch, Sadiq & Fabritiis, 2011*) or use brute force equilibrium approaches (*Giorgino, Buch & Fabritiis, 2012*) to improve resolution.

For future studies, although our approach offers the simplicity of not requiring prior knowledge about a system of interest (other than a bound model), at this point SMD may not the best approach for many relative affinity calculations. To ensure one's results are independent of the dissociation path one selects would require computing the work of separation for all likely paths. Such an approach eventually requires using the Jarzynski inequality (*Jarzynski, 1997*) to establish a lower limit for binding energy and would quickly become computationally inefficient for evaluating a large number of mutations in most systems. However, considering the strong correlation between FEP and SMD in this system, it may not be important to ensure one's results are path independent for relative affinity calculations, as long as the same path is used for all complexes.

More importantly, with no *a priori* knowledge of the appropriate number of equilibration samples, the best duration of equilibration, the appropriate number of pulling runs, or the best pulling speed means the computational expense in our SMD protocol may not be commensurate with the information provided. For example, another all atom approach that makes calculations via short simulations of spatially restrained complexes has proven capable of generating relatively accurate binding affinities with less compute time than is required from our steering strategy (*Gumbart, Roux & Chipot, 2013a*; *Gumbart, Roux & Chipot, 2013b*). That being said, there is no reason to believe this SMD approach to mutagenic studies could not be optimized to reduce computational expense. Further analysis will be needed to understand the lower limits of resources required for accurate predictions.

## ACKNOWLEDGEMENTS

This work was carried out using high-performance computing resources provided by the High Performance Computing Center (HPCC) at Texas Tech University at Lubbock (http://www.hpcc.ttu.edu) and the Texas Advanced Computing Center (TACC) at The University of Texas at Austin (http://www.tacc.utexas.edu). We would like to thank Bryan Sutton for opening access to the Hrothgar cluster and the reviewers Ilan Samish and Matteo Masetti for their helpful comments on this work.

### Funding

This work was supported by the Defense Threat Reduction Agency (HDTRA1-12-C-0007) to ADE, SLS, and COW, the National Science Foundation (MCB-0943383) and the Welch Foundation (F-1654) to ADE, the National Institutes of Health (R01-GM088344) to COW, and the National Institutes of Health (R01-GM093086) to SLS. The funders had no role

in study design, data collection and analysis, decision to publish, or preparation of the manuscript.

## Grant Disclosures

The following grant information was disclosed by the authors:

Defense Threat Reduction Agency: HDTRA1-12-C-0007.

National Science Foundation: MCB-0943383.

Welch Foundation: F-1654.

National Institutes of Health: R01-GM088344, R01-GM093086.

## Competing Interests

The authors declare no competing financial interest.

## Author Contributions

- Austin G. Meyer conceived and designed the experiments, performed the experiments, analyzed the data, contributed reagents/materials/analysis tools, wrote the paper, prepared figures and/or tables, reviewed drafts of the paper.
- Sara L. Sawyer and Andrew D. Ellington conceived and designed the experiments, reviewed drafts of the paper.
- Claus O. Wilke conceived and designed the experiments, analyzed the data, wrote the paper, reviewed drafts of the paper.

## Supplemental Information

Supplemental information for this article can be found online at http://dx.doi.org/10.7717/peerj.266.

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
