# Peer review of "Analyzing machupo virus-receptor binding by molecular dynamics simulations"

_PeerJ, doi:10.7717/peerj.266_

## Round 0.1 · original submission · Major Revisions

There are two issues in my view that need to be addressed by the authors.

The first issue, raised by both reviewers, is that of adding more details. Both reviewers as for clarifications in several points and also the use of more precise language (even in the title). The reviewers point to several places in the manuscript where such changes are necessary and in my view this will require some effort in rewriting the manuscript.

The second issue that needs to be addressed is that raised by reviewer 2 on the validity of the approach in relation to the results shown for the effect of mutations. The reviewer suggests two approaches but I leave it to the authors to argument the merits or pitfalls of these suggestions and follow up on them or suggest yet a different alternative to address this point.

·

Basic reporting

The authors use steered molecular dynamics to separate the machupo virus (MACV) spike glycoprotein (GP1) away from the human transferrin receptor (hTfR1) and estimate affinity, mutation effect and role of protein-protein interaction (PPI) H-bond networks. Overall, the results are promising and the level of accuracy is high.

1. I don't know the word limit of the tilte but the journal editor should consider changing the paper's editor to reflect the fact that a single case-study (GPI-hTfR1) is analyzed rather than the general phenomenon of virus-receptor binding. This is needed especially as different case-studies often produce different results. Indeed, the authors well articulate already in the abstract and then in the introduction the weak spots of the field. In the same line of thought, the case-study differences and the analyzed data is not only b/c of PPI problems. An important paper in the field that caused David Baker to work hard on the problem is http://www.ncbi.nlm.nih.gov/pubmed/?term=19561092 where it shown the mutation prediction is good on average but not in each case-study. Actually, the authors may want to refer to this paper in their mutation analysis. Likewise, steered MD was already used for PPI e.g. Cuendet M, Michielin O. 2008 which the authors cite.

2. The authors imply that their methodology is new. Yet, steered MD is a common methods and was used also for mutaions e.g. pubmed I (PMID: 23836247 ). The relevant statement should be softened.
.

Experimental design

1. The spring constant was set to 5 kcal/mol/ ˚ A and the velocity set to 1 A/ns. This choice of parameters should be discussed in light of other SMD publications as are the other choices within the SMD protocol. In this discussion, differences should be highlighted and explained.

2. in section 3.2 you write: "For SMD, at the centroid of 176
several atoms a force was applied to a single point in GP1." Readers should be able to generally repeat the simulation and actually the choice of the atoms to be pulled may affect results. Thus, the list of pulled atoms should be attached as a supp material.

Validity of the findings

The good correlation of results to experiment and the detailed logical explanation of mutation effect is impressive.

Additional comments

Minor comment -it would be nice (but not must) to change table 3-5 - highlight the number differences visually by a color specture.

Overall, the results are interesting and the analysis seems sound but the protocol should be better presented and explained in light of other SMD studied, especially as the field suffers from too much over-analsyis of the trajectory and wrong choice of parameters.

·

Basic reporting

The paper is interesting and well written.

Experimental design

Methods are not enough detailed.
Why did the authors use a steering force acting on all alpha carbon instead of the more common choice of either steering just one atom or COM of one of the two counterparts? Are the authors using the protocol introduced by Cuendet and Michielin?
If so, this must be clearly stated, otherwise much more details should be reported in the Methods section regarding the steering procedure.
Moreover, certain methodological details are reported in the Results (see page 8, rows 176, 177), increasing the difficulty in understanding the steered MD procedure.

Validity of the findings

Results are not as relevant as they should be, and the analysis is not sufficiently detailed.

1. In the Results, at page 7 (rows 140 to 143), the authors state that large structural rearrangements of an interfacial flexible loop are expected upon binding, and they claim that because of this traditional computational tools are of limited usefulness. While one might agree with this sentence, the authors seem to overlook the fact that such a rearrangement complicate any other simulative strategy as well. How does this loop behave during steered MD simulations?
Does it unfolds or it stays in place?
A more detailed structural analysis is required.

2. As a follow up of the previous point, where are the mutations located relatively to the protein-protein interface?
Are they located in the flexible loop? In this case, how do the complexes "react" to the mutations?
Are there major rearrangements or just local motions during the production runs preceeding the pulling phase?
Very basic analysis such as computing the RMSD over time and RMSF per residue should be performed and reported.
Moreover, showing the location of these mutations in Figure 1 would improve the comprehension of the manuscript.

3. I acknowledge the authors that probing the viability of the approach to point mutations before moving to more challenging cases is a good practice, and I also appreciate the fact that they do not overemphasize their results.
However, if I properly understood, the only experimental validation of the protocol is provided by three point mutations which returned two statistically insignificant correct results (N348K, vR111A) and one statistically significant wrong result (Y211T).
Unfortunately, this is not enough to support the steered MD protocol.
I suggest the authors to enrich the paper in either one of the two following ways:
1. by performing a detailed comparative analysis of the interactions and their changes upon mutations along the steereing phase (both from a structural and energetic point of view, e.g. by calculating the interaction energies as reported in the Cuendet and Michielin paper, by monitoring hydrogen bonds braking/formation and so on),
OR
2. to compare the steered MD results for the three alanine point mutations with more established free energy approaches (FEP or TI, see for example the recent paper by Luitz and Zacharias, Proteins 2013 81:461).

Additional comments

There are conceptual points that need to be addressed in the current form of the manuscript.

1. In the Introduction, at page 1, (rows 14 to 17), the authors state that methods such as FEP and TI rely on a two state model, "with no intermediate steps". This is in a best case scenario a misleading sentence, since FEP and TI are computational frameworks that, in order to return meaningful results, in most of cases DO NEED intermediate steps. In fact, the sentence is also misleading since the authors refer to these methods as to "two state models" whereas they are generally known as "pathway methods" to be easily distinguished by "endpoint methods" such as MM-PBSA, LIE, ..., where only ensemble averages of certain endpoint properties are used without intermediate states!
This source of confusion must be properly addressed.

2. In the Introduction, at page 1 (row 20), the authors refer to the use of steered MD in the prediction of relative binding affinity between wild type and mutant protein-protein complexes as a "new method".
This is an overrated sentence for two reasons: 1. the authors show a procedure, a protocol (if it was properly detailed), they do not develop any new methods; 2. a similar approach has already been reported
by Cuendet and Michielin as early as 2008, as the authors properly acknowledge in the Results section.
The sentence should be tuned down, and this also holds for the sentence at page 11 (rows 266, 267) where the auhtors state that they changed the scope of application of SMD simulations.

Minor points.
1. Page 1, row 14: probably a typo in: "...to forgo training."

2. Page 4, row 71. Is the system size really of about 28,000 atoms? Systems of about 30,000 atoms are relatively small, whereas by looking at Figure 1B I would have expected at least 50,000 atoms. Please double check.

3. Page 4, rows 74, 75: "This was accomplished by setting beta=1." What do the authors mean? What does the beta parameter stand for?

4. Page 4, row 84. The authors state that the exact input file is reported in the supplement. I wasn't able to find any input file in the supplement, but I apologize beforehand with the authors in case the input file was actually provided.
Please double check. Anyway, the authors should at least indicate in the main text very important simulation parameters such as the temperature control method employed and the cutoff radii for nonbonded interactions.

5. Page 6, rows 131, 132. Please provide some more details regarding the "pairwise.t.test function" and the "FDR p-value correction".

6. Page 8, row 171. Please avoid odd sentences such as "we generated a physically realistic simulation world", as this is what people usually do when performing Molecular Simulations, there is no need for such an emphasys in a scientific paper.

7. It seems to me that the authors have a menageable amount of data. Just showing a force profile (Fig. 2) and the maximum force probability distribution (Fig. 3) for selected entries in the main text is ok, but the authors should also show all the force profiles and maximum force probability distributions for all the system they simulated in the supplement.

8. Page 13, row 321. What do the auhtors mean with "unrestrained SMD"?

9. Following the same order of the muations listed in Table 1-5 would improve the readibility of the Results.

---

## Round 0.2 · Minor Revisions

The revised manuscript covers all issues raised by the original reviewers. I would you to take a look at the minor issues raised by Reviewer 1, particularly clarifying figure 1 with the extra detail. Also citing the relevant reference and while at it, why not standardizing the amino acid nomenclature?

·

Basic reporting

Remark 1:

In a report appearing in the in June 2013, We used SMD methodology to study the structural stability the interface of the Trp-repressor using WT dimeric form as reference comparison with mutant cases. Those results did show significant differences and consistent with experimental outcomes. The criteria for the establishment of those differences were force-extension profiles and areas under curves in similar fashion of this actual report of Meyer et al.

Due adherence to scientific standards requires the recognition of the fact mentioned above and the respective citation of our work:

Miño G, Baez M, Gutierrez G. Effect of mutation at the interface of Trp-repressor dimeric protein: a steered molecular dynamics simulation. Eur Biophys J. 2013 Sep;42(9):683-90

Remark 2:

An image, probably a C) part of figure 1, showing the structural details of the key analysed residues and also depicting the two large hydrogen-bonding networks in the GP1/hTfR1 interface is needed in order to facilitate the reading and understanding of the main results of the research.

Remark 3:

The narrative make use of two codes for aminoacid labelling, one of one letter (in tables) and the other of three letters (In the text). I suggest to use only one of them for text and tables.

Remark 4:

In line 201 a Typo appears : ...by using the smooth.spline() and predict() functions...

Experimental design

No Comments

Validity of the findings

No Comments

Additional comments

No Comments

·

Basic reporting

no comments

Experimental design

no comments

Validity of the findings

no comments

Additional comments

The authors have answered and clarified all the points in question - the ms should be accepted.
Note to the editor / PeerJ staff: the supporting material currently deposited on an external site https://github.com/clauswilke/MACV_SMD should be deposited as SI on the PeerJ site.

·

Basic reporting

No Comments

Experimental design

No Comments

Validity of the findings

No Comments

---

## Round 0.3 · accepted · Accept

All issues have been resolved.